# Failure mechanism of soft– hard-interbedded rock slopes in cold regions: Numerical simulation and theoretical analysis

**Lilong Ma**[1], **Runsen Lai**[2], **Zizhao Zhang**[2,3]*, **Yanyang Zhang**[2], **Qianli Lv**[2], **Guangming Shi**[2], **Junpeng Huang**[2]

**1** Xinjiang Uygur Autonomous Region Geological and Mineral Exploration and Development Bureau, Third Geological Brigade, Korla, China, **2** School of Geology and Mining Engineering, Xinjiang University, Urumqi, China, **3** Research Base of Xinjiang University, State Key Laboratory of Intelligent Construction and Healthy Operation and Maintenance of Deep Underground Engineering, Urumq, China

☉ These authors contributed equally to this work.

* zhangzizhao@xju.edu.cn

## Abstract

The soft– hard- interbedded rock slope in cold regions generally undergo the differential weathering due to the freeze-thaw effects, for which the irregular rock fractures increase the risk of geological disasters occurrence. To investigate the failure mechanism of the rock slope, both the numerical simulation and theoretical analysis were adopted in the present research. The structural integrity of the soft– hard- interbedded rock slope subjected to three different conditions (e.g., freeze-thaw cycles, natural, and anchored surroundings) was evaluated. The results indicated that the stability of the rock slope is significantly affected by the freeze-thaw cycles, the performance of which is much different from that under the natural- and anchored conditions. The failure process of the soft– hard- interbedded rock slope in cold regions exhibited the unloading-tensile cracking-sliding characteristics. The damage of the irregular structural surfaces subjected to the long-term weathering and freeze-thaw cycles is the one of the main factor controlling the failure mode. As revealed by numerical simulation, the application of the anchor rods can effectively prevent and control the collapse of the soft– hard- interbedded rock slope. The findings obtained from this research provide an important guidance for the stability assessment and mitigation design of the soft– hard- interbedded rock slopes in cold regions.

## 1. Introduction

The structural integrity of the soft-hard- interbedded rock slope is always a critical issue to be accounted for, in particular, when it is subjected to the freeze-thaw cycles in cold regions. Affected by the repeated change of natural climate, the rock masses usually suffered from somehow strength reduction associated. In general, the weathering resistance of the hard rock seems to be stronger than that of the soft rock, the latter of which will show the unexpected

**Data availability statement:** All relevant data are within the manuscript and its Supporting Information files.

**Funding:** This work was supported by the following 2 funds, programs, projects: Scientific Research Programme "Tianshan Excellence":(2023TSYCCX0010), The National Natural Science Foundation of China (42367021). The funders is mainly responsible for study design, data collection and analysis, and decision to publish.

**Competing interests:** The authors have declared that no competing interests exist.

spalling and falling off under the critical climate. The progressive strength reduction is believed to be the main reason resulted in geological disasters (e.g., the landslides and the collapses) [1]. During past several decades, the rapid development of the modern cities promotes the expansion of infrastructure projects (i.e., the highways, railways, as well as the water conservancy projects) and several geological disasters consequently occurred. Among these engineering projects, the integrity of the soft–hard- interbedded rock slope has drawn much attention attributed to its complex failure mechanism and vital threaten to local citizens [2].

It has been well noted that the failure mechanism of the soft–hard- interbedded rock slope is predominately controlled by the interface between different rock layers, which is also the weakest zone to be accounted for [3]. Hard rocks are typically of high strength and large stiffness, while soft rocks exhibit the lower strength associated with the large deformation ability. In this case, the stress concentration generally occurred around the slip surfaces, once the external force was applied on the soft–hard- interbedded rock slope. Apart from the natural mechanical properties of the rock mass, the change of the exterior surroundings also affects the structural integrity of the soft–hard- interbedded rock slope. For example, a large amount of reported landslides are related to the freeze-thaw cycles, rainfall, as well as the groundwater, when other parameters are the constant. In fact, the climate changes will not only exacerbate the fracture of the soft rock, but also weakens the hard rock, resulting in the accelerated instability of the rock slopes. As previously demonstrated by E. Hoek et al [4],the failure mode of the rock slope can be classified into four groups: the planar sliding, the circular sliding, the wedge sliding, and the toppling failure. Furthermore, it is Stead D. et al who systematically investigated the impact of geological structures and divided the failure modes of rock slopes into eight types [5]. Upon the laboratory tests, Zhao et al found that the progressive deformation process of the rock slopes consists of three stages: the initial unloading rebound, the long-term gravity bending, and the late-stage creeping state [6]. Compared to physical tests, the application of the numerical modelling offers the new insight into the analysis of engineering problems with complex conditions [6–15]. The numerical simulations make it possible to visually capture the progressive changes of the stress state and the failure process of the slopes under variable conditions. It is Yao et al who evaluated the stability of the soft–hard- interbedded rock slope via the inherent strength reduction programme in the FLAC3D, the critical parameters investigated in which covered the dip angles, the angles between rock layers, as well as the slope direction [16]. Different from the failure mechanism of the toe slide on the inclined table [17], Tan et al numerically evaluated the influence of rock mass structure on the mining slope with the discontinuous deformation analysis method [18]. Afterwards, the UDEC program was also adopted to explore the failure mechanism and the deformation process of rock slopes with variable joints [19,20]. Some other scholars further evaluated the stability of the anti-dip rock slopes with the soft-hard inter layer with the combination of the UDEC program and the orthogonal experiment [21]. Compared to physical investigations, the application of the UDEC software to simulate hazardous slopes is not only much more accurate but also cost-effective. Meanwhile, only the rock discount rate should be obtained to simulate the rock slopes under different working conditions.

The aforementioned discussion demonstrates the feasibility to explore the failure mechanism of the soft–hard- interbedded rock slope through the numerical simulation and theoretical analysis. In the present research, the basic numerical model for the soft–hard- interbedded rock slope at Camel Peak in the Altay region of Xinjiang was established, followed by the detailed analysis on the deformation characteristics of the slope under the natural condition. The failure mechanism of the soft–hard- interbedded rock slopes subjected to the freeze-thaw cycles and anchoring conditions were then investigated. The research outcomes will contribute to the in-depth understanding about the failure mechanism of the soft–hard- interbedded

rock slope, but also provide a scientific reference for the stability assessment of the rock slopes with similar geological conditions.

## 2. Numerical modelling and condition settings

### 2.1. Establishment of the basic model

The UDEC program developed by the ITASCA International was applied in the present research. As a typical discrete element analysis program, it is ideal to simulate and calculate the large deformation of rock masses. Attributed to its superior advantages, the UDEC can effectively simulate the physical and mechanical behavior of discontinuous medias either under the static or the dynamic loading conditions [22]. Herein, the UDEC–Voronoi model was adopted to evaluate the slope stability in this section [23–25]. Based on the field investigations [26], the two-dimensional slope calculation model with the typical cross-section shown in Fig 1 was established.

Considering that the main aim of this research is to explore the failure mechanism of the soft– hard- interbedded rock slope, the basalt was selected to represent the hard rock, while the tuff was adopted as the prototype for the soft rock. As illustrated in Fig 1, the dimension of the numerical model ranges from 300 to 400 meters. Note that the both the horizontal and vertical displacement of the bottom boundary is constrained, while the upper and open surfaces of the slope are left as the free boundaries. The Voronoi polygon with a configuration of 10 meters was meshed and the Mohr-Coulomb brittle fracture constitutive model was adopted to generate the virtual interface. For ease to capture the progressive deformation of the rock slope, there are three monitoring points (i.e., 1#, 2#, 3#) located at the top and middle of the numerical model.

### 2.2. Determination of properties of rock mass

The mechanical properties of the rock mass were obtained from the field sampling and laboratory testing. The work of this part has been completed in the early stage, which can be seen in Reference [27]. Since that the rock slope is the combination of the inter-layered soft and hard rocks, as listed in Table 1, different physical and mechanical parameters values were assigned to the model, respectively. To simulate the actual collapse affected by the freeze-thaw cycles which mainly occur during the transitional period between the spring and winter [28], the number of the freeze-thaw cycles were determined to 30 times for deterioration calculations.

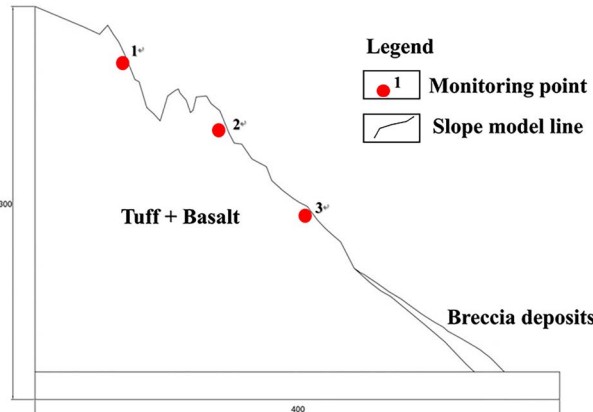

**Fig 1.  Slope calculation model (unit m).**

**Table 1. Physical and mechanical parameters of rock and soil under natural conditions.**

| Material name | Density/Unit weight (kg/m³) | Bulk modulus (MPa) | Shear modulus (MPa) | Cohesion (MPa) | Tensile strength (MPa) | Internal friction angle (°) | Elastic modulus (GPa) | Poisson's ratio | Uniaxial compressive strength (MPa) |
|---|---|---|---|---|---|---|---|---|---|
| Tuff + Basalt | 2786 | 51 | 40 | 1.27 | 6.55 | 46 | 0.69 | 0.19 | 65.75 |
| Breccia deposits | 1900 | – | – | 0.02 | – | 40 | – | – | – |

**Table 2. Physical and mechanical parameters of rock and soil under freeze-thaw conditions.**

| Material name | Density/Unit weight (kg/m³) | Bulk modulus (MPa) | Shear modulus (MPa) | Cohesion (MPa) | Tensile strength (MPa) | Internal friction angle (°) | Elastic modulus (GPa) | Poisson's ratio | Uniaxial compressive strength |
|---|---|---|---|---|---|---|---|---|---|
| Tuff + Basalt | 2786 | 37.20 | 29.17 | 0.93 | 4.78 | 33.55 | 0.50 | 0.19 | 41.35 |
| Breccia deposits | 1900 | – | – | 0.014 | – | 29.17 | – | – | – |

Considering that the maximum frozen soil depth in the research area is about 1.6 meters and thus rock samples below this depth were sampled [29,30,31].

It can be seen from Table 2 that the uniaxail compressive strength of the rock samples decreased from 65.75 MPa to 41.35 MPa after 30 freeze-thaw cycles. Compared to the initial uniaxial compressive strength, the strength deterioration rate is about 37.11%. As per this experimental observation, the strength reduction method was adopted in the subsequent numerical simulation with a constant deterioration rate discussed herein.

The simulation mainly uses the FISH language in the UDEC software to import the boundary of the slope calculation model, and then through the 'block create', 'block group', and other commands to calculate the established model. After that, we can use 'block create', 'block group', and other commands to fill the blocks, which are mainly filled or grouped by Tyson polygon mesh; and then we can use 'block prop mat', 'prop mat', 'block contact' and other commands to fill the blocks. The blocks are assigned values by 'block prop mat', 'prop mat', 'block contact', and other commands, and finally, the stability of slopes under different working conditions is calculated.

## 3. Failure mechanism of the soft–hard- interbedded rock slope

### 3.1. General observation

**3.1.1. Rock Slopes under the natural condition.** As can be seen from the displacement cloud diagrams of the rock slopes shown in Figs 2 and 3, some cracks appear in the unstable rock mass at the top of the slope due to gravity when it is under the natural condition. Even though, the overall slope can still remain the relatively stable and only some local detachment was observed at the foot of the slope. The stress cloud diagrams depicted in Figs 4 and 5 indicate that inherent stress within the slope is primarily concentrated at the top and bottom of the slope, with a significant stress concentration observed at the base of the slope.

**3.1.2. Rock slopes under the freeze-thaw cycles.** Under the freeze-thaw conditions, significant displacement occurs in the unstable rock mass at the top of the slope, which can be seen from Figs 6 and 7. The occurence of cracks between rock blocks is mainly attributed to the combined effects of freeze-thaw cycles, the weathering, as well as the gravity. The other observation shown in these diagrams is that although noticeable rock falls at the top associated with local cracks, the middle section of the rock slope remains stable. At the base, breccia deposits cause the soil to bulge slightly due to gravitational forces. It is apparent in Figs 8 and 9 that the stress distribution under the freeze-thaw conditions are much diffeent that under the natural condition. That is, the stress primarily concentrated at the top and bottom of the slope, while a significant concentration was observed at the base.

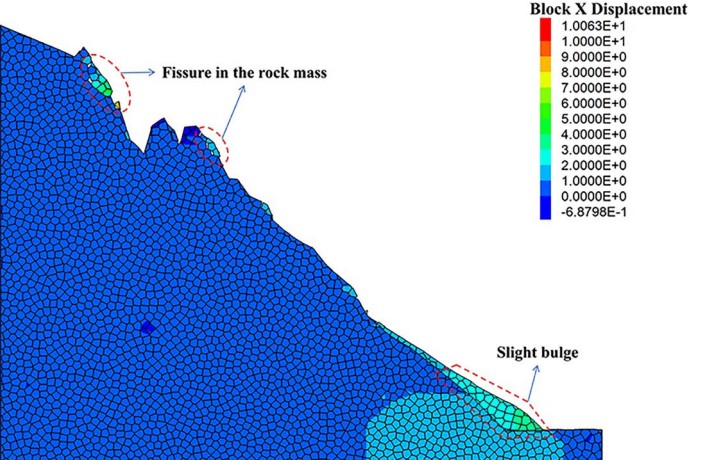

**Fig 2. X-direction displacement cloud diagram of natural condition model.**

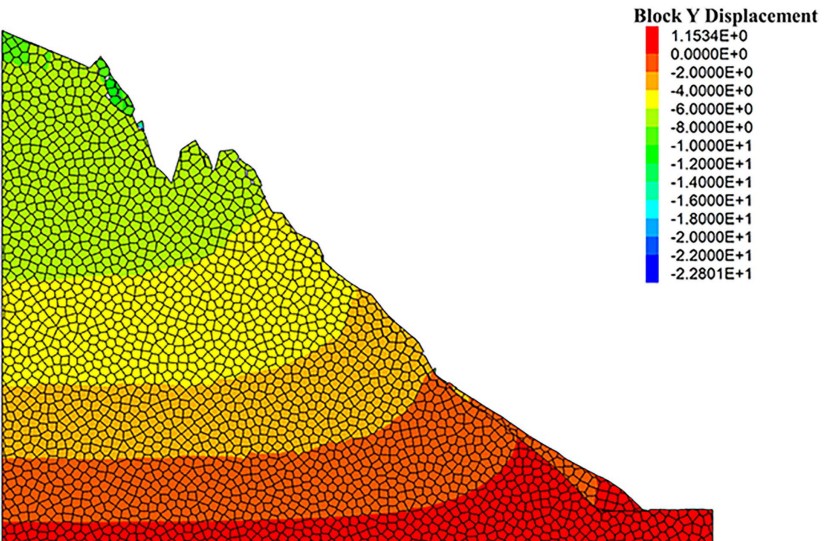

**Fig 3. Y-direction displacement cloud diagram of natural condition model.**

## 3.2. Progressive deformation of the slope

**3.2.1. Characteristics of the displacement.** To monitor the collapse of the rock slope under the natural condition, time series curves of Y-direction and X-direction displacement changes at each monitoring point were analyzed. As depicted in Fig 10a, the X-direction displacement remains relatively small, with a maximum of only 0.82 mm. Among them, the smallest degree of change in x-direction displacement was capatured by the 2# monitoring point. In contrast, significant Y-direction displacement is observed at 1#, 2#monitoring points, with a maximum displacement of -8.19 mm. The simulation results indicate that the rock slope exhibits the substantial Y-direction displacement and minor X-direction displacement. It is clear that the Y-direction displacement is mainly negative (Fig 10b), while X-direction displacement is mostly positive. Because that the rock and soil mass in the unstable rock area

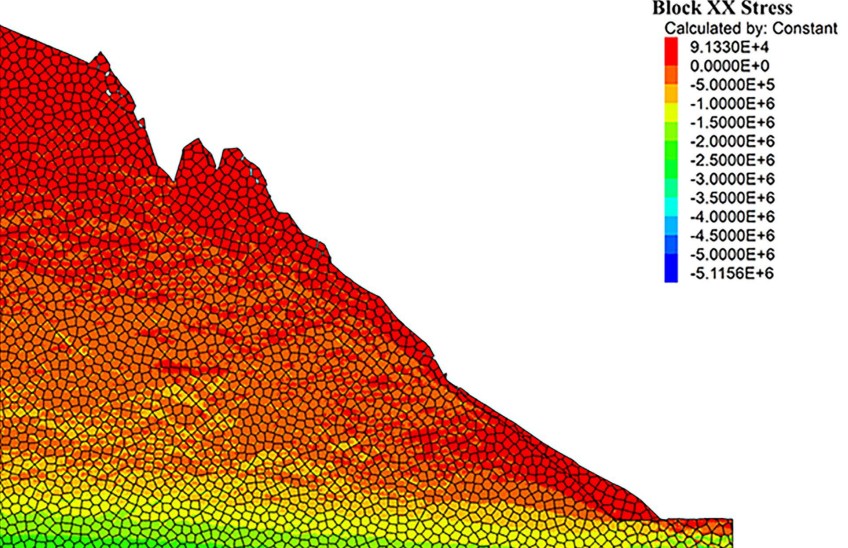

**Fig 4. X-direction stress cloud diagram of natural condition model.**

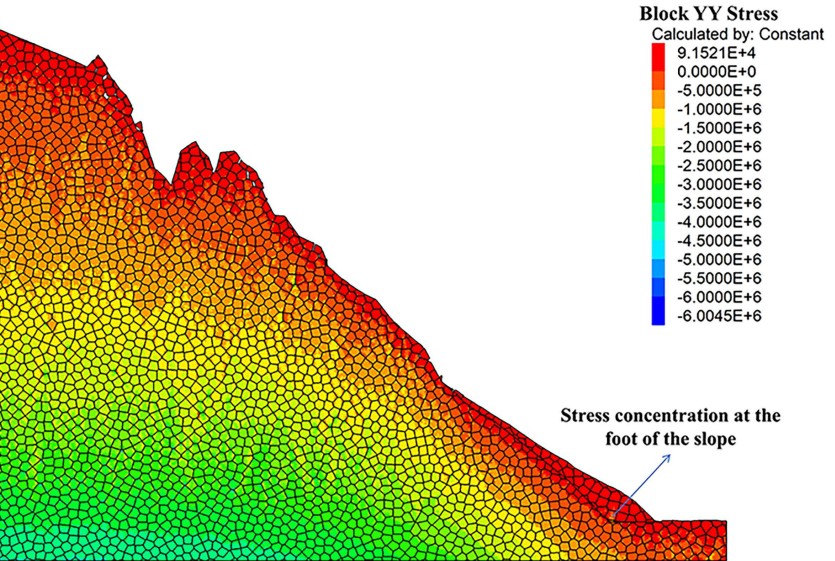

**Fig 5. Y-direction stress cloud diagram of natural condition model.**

undergo cracking or detachment, there is an uplift at the foot of the slope where the breccia deposits are located.

As depicted in Fig 11a and 11b, the collapse under the freeze-thaw condition reveals that the X-direction displacement is relatively small, with a maximum displacement of 0.92 mm. It can be seen from the data obtained by 2# Monitoring point that only minimal changes in X-direction displacement during the rock and soil simulation. However, significant Y-direction displacement is observed at 1 # and 2# monitoring points on the slope with the displacement of -5.6 mm. Compared to the simulation results under natural conditions, the Y-direction displacement under freeze-thaw conditions is smaller. This reduction is

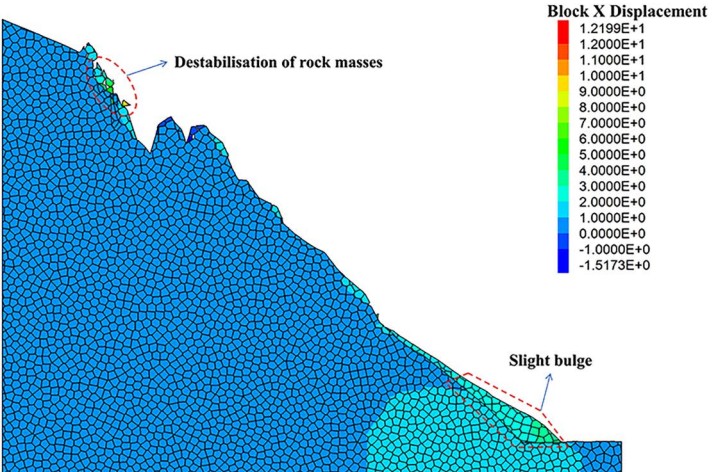

**Fig 6. X-direction displacement cloud diagram of freeze-thaw condition model.**

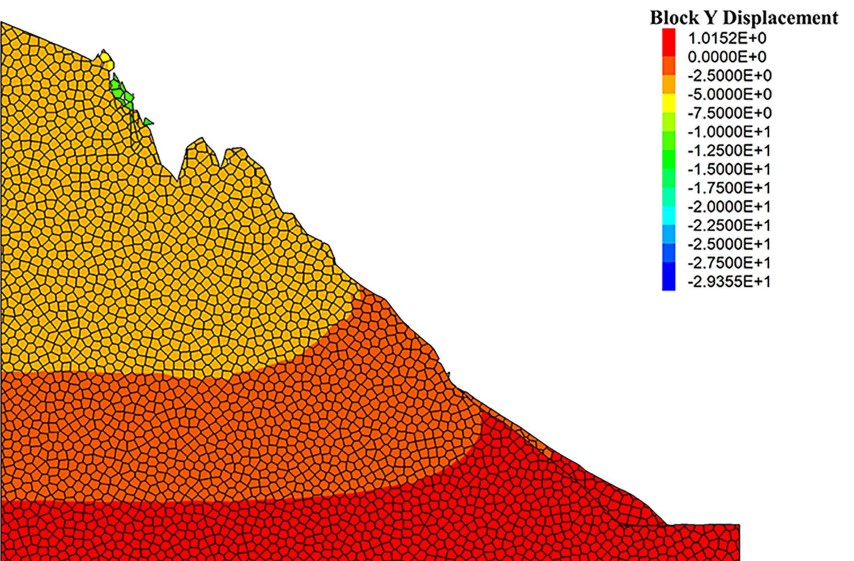

**Fig 7. Y-direction displacement cloud diagram of freeze-thaw condition model.**

mainly because the unstable rock mass has already experienced instability and failure, as indicated by noticeable rock falls in the diagram. The Y-direction displacement is primarily negative, with uplift occurring in the middle section of the slope.

**3.2.2. Characteristics of the stress.** Discrete element numerical simulations were performed on the slope model of the study area to simulate and monitor stress distribution under the natural condition. As illustrated in Fig 12a and 12b, the stress variation trends at each monitoring point were analyzed through the time series curves. Herein, the positive and negative values are adopted to represent the direction of stress. The stress changes in the X-direction are relatively stable (Fig 12a). From the initial stage of stress fluctuation to the equilibrium process of stress, the stress measured by each monitoring point shows a decreasing trend in value. However, the change of stress shows an overall increasing trend

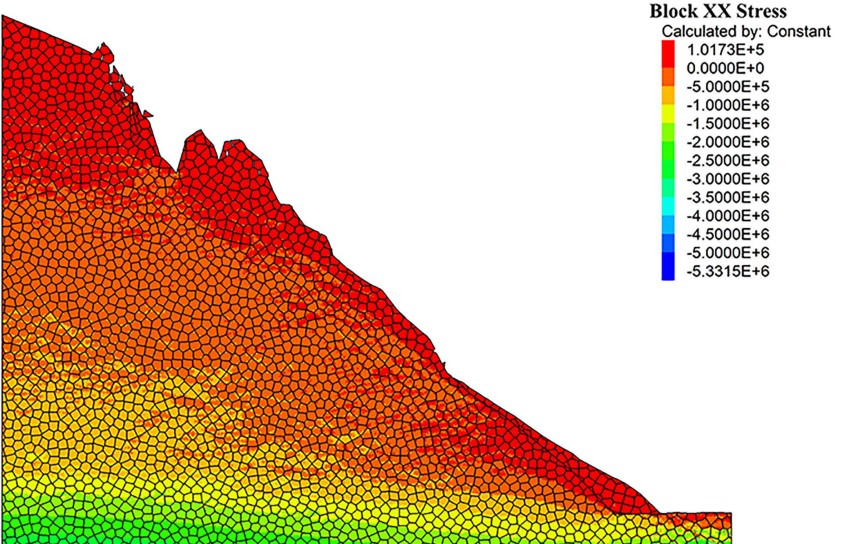

**Fig 8. X-direction stress cloud diagram of freeze-thaw condition model.**

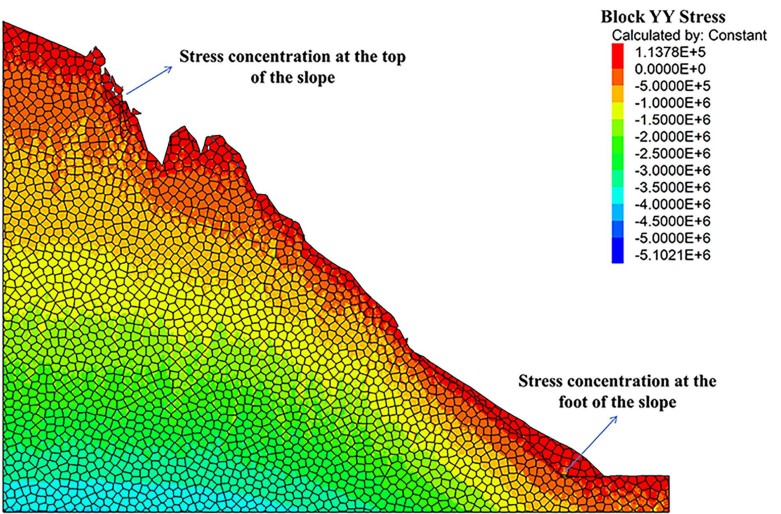

**Fig 9. Y-direction stress cloud diagram of freeze-thaw condition model.**

from the Y-direction. The larger increase obtained from 1# monitoring point 1 suggests that cracks at the slope top influence the stress distribution, though the overall stress changes are relatively small during the whole process.

Fig 13a and 13b shows the tress changes when the slope is under the freeze-thaw condition, in which the stress process recorded by 1# monitoring point can be divided into three stages. The first stage is the initial stage, where the stress curve gradually increases. The second stage is the stability stage, during which the stress curve gradually tends to equilibrium. The third stage occurs when the rock body occurs damage, and the stress enters into the destructive stage. Within this stage, the stress curve appears to be steeply inclined decline, and the continuous decline until the slope is stabilized. This occurrence of this pattern is mainly attributed

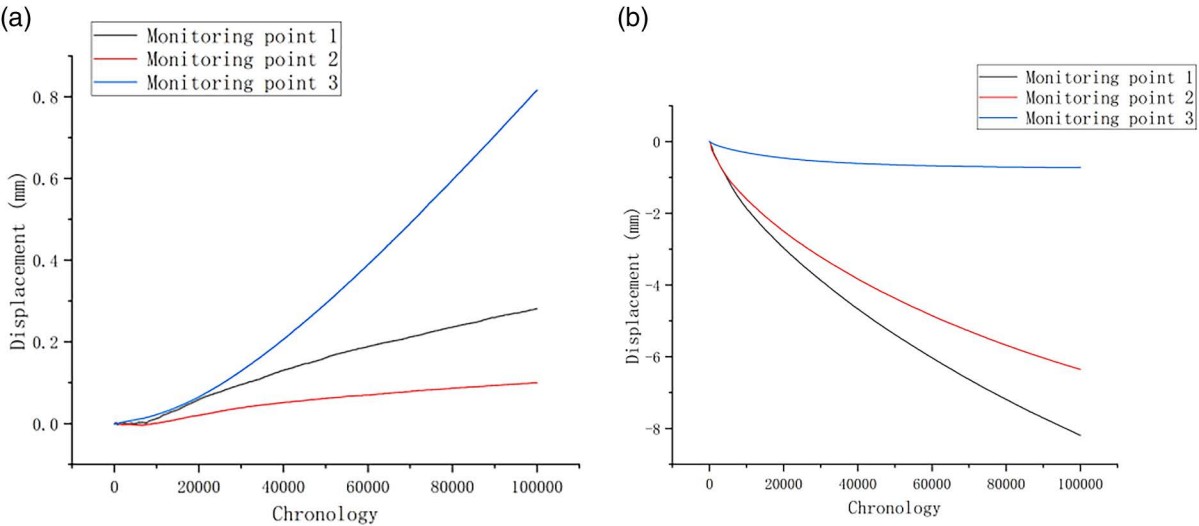

**Fig 10. (a)** X-direction displacement statistics of monitoring points under natural conditions; **(b)** Y-direction displacement statistics of monitoring points under natural conditions.

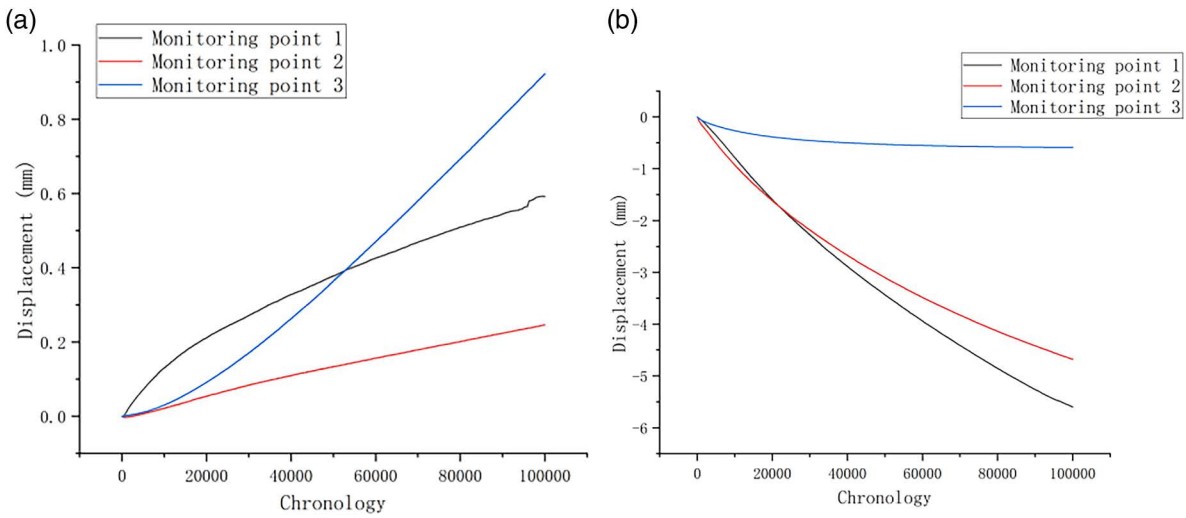

**Fig 11. (a)** X-direction displacement statistics of monitoring points under freeze-thaw conditions; **(b)** Y-direction displacement statistics of monitoring points under freeze-thaw conditions.

to the formation of fissures, destabilization, and the breaking and restabilization of the rock structure. the similar observation also can be seen from 2# and 3# monitoring points, which experience an initial increase in stress that gradually stabilizes, with stress changes being relatively concentrated on the slope surface.

Based on the UDEC model simulation results, the slope remains stable under natural conditions, with displacement primarily concentrated in the upper part of the slope. However, under freeze-thaw conditions, the upper part of the slope becomes unstable, leading to instability in the middle section as well. This instability can cause localized rock falls, resulting in gravel accumulating at the slope's base due to self-weight in both conditions.

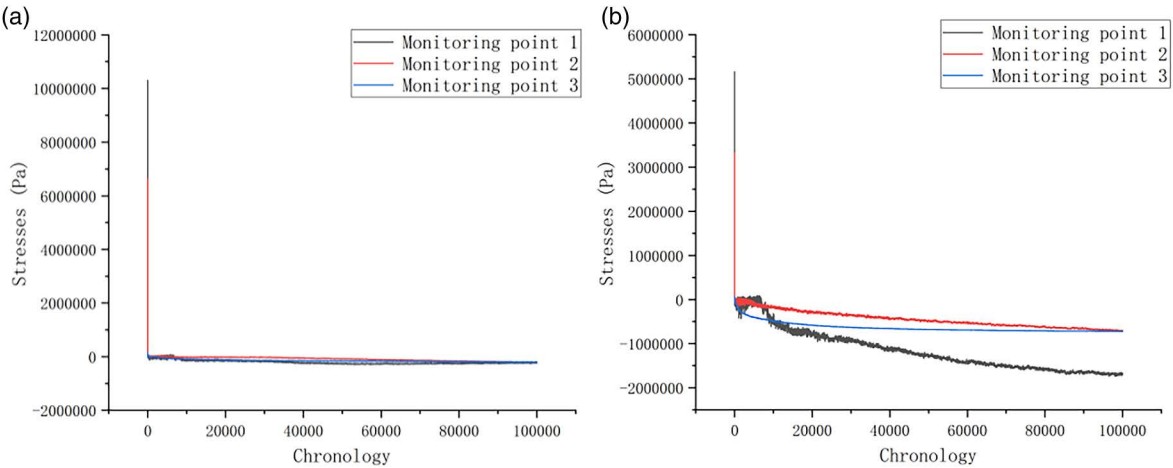

**Fig 12. (a)** X-direction stress statistics of monitoring points under natural conditions; **(b)** Y-direction stress statistics of monitoring points under natural conditions.

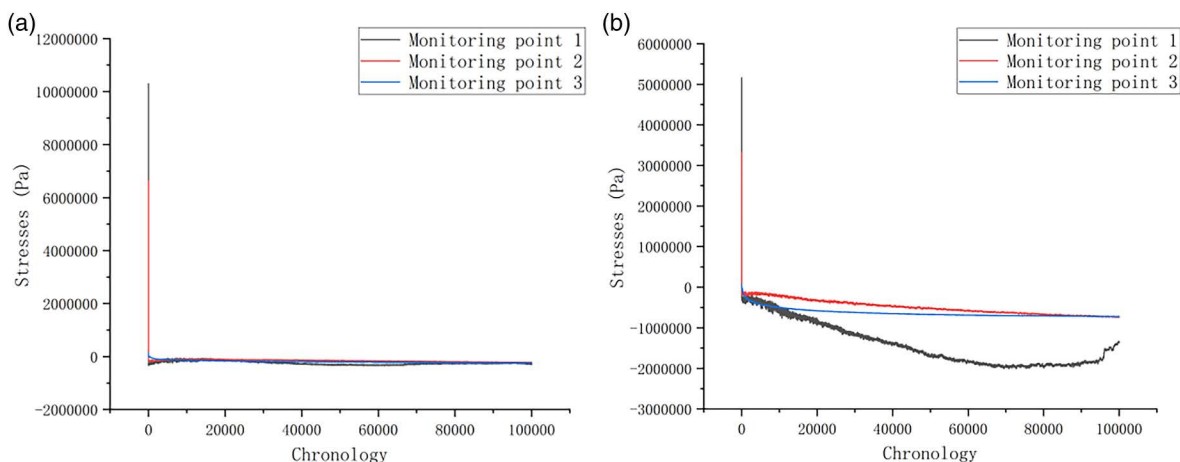

**Fig 13. (a)** X-direction Stress statistics of monitoring points under freeze-thaw conditions; **(b)** Y-direction Stress statistics of monitoring points under freeze-thaw conditions.

## 3.3. Failure mechanics of rock slopes

**3.3.1. Topographical conditions.** The rock mass of Camel Peak exhibits well-developed small and medium-sized structures, including small folds, cleavage, and joints. The slope features 16 unstable rock masses with height differences ranging from 20 to 120 meters. These conditions create a dynamic environment conducive to the destabilization of these hazardous rock bodies, posing a risk of impacting the road below after the collapses.

**3.3.2. Rock mass structure.** The joints and cracks in the rocks are grouped, with the typical pattern at the Camel Peak collapse being an "X" joint combination. This combination disrupts the original stress field of the rock mass, leading to local stress superposition and the formation of through-going joint fracture surfaces. The inclination angles of these cleavage surfaces generally ranges from 30° to 60°, while the values of whcih are larger than 60°. The mixed lithology of tuff and basalt results in differential weathering, creating irregular and staggered fissures, forming the characteristic "X"-type joints.

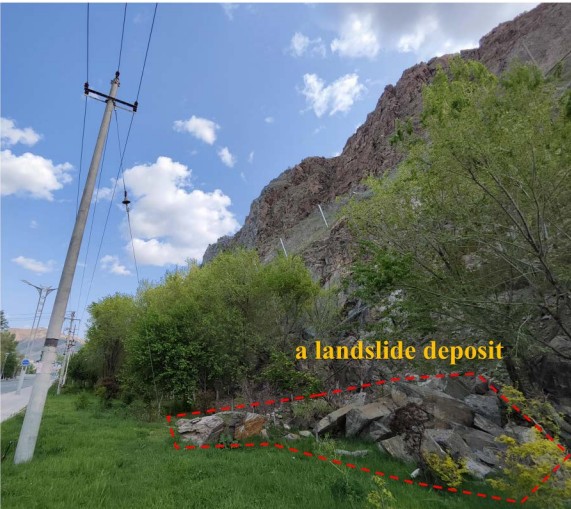

**Fig 14. Rockfall accumulation in front of slope.**

**3.3.3. Inducing factors.** It is in the Altay region where frequent and cyclic freeze-thaw processes damage the joints and cracks in the rock. These cracks are mainly featured with the "X" shaped shear failures, which results in irregular and unstable structural surfaces. Moreover, long-term weathering and freeze-thaw actions will further weaken the hazardous rock mass. Attributed to its own weight along the structural surface and the natural voids created by the differing properties of hard and soft rocks, the collapse of rock slopes is easily triggled. The resulting rock and debris from these avalanches are typically large in volume, moving primarily by rolling and jumping to the base of the slope (see Fig 14). During this process, falling rocks may strike other destabilized rocks on the slope, triggering additional rock slides.

**3.3.4. Collapse failure mechanism.** Based on the UDEC numerical simulations and analysis of various factors, as depicted in Fig 15, the collapse mechanism of the soft– hard-interbedded rock slope can be summarized as follows. Geological tectonics create a soft and hard mixed rocky slope with a significant elevation difference. Differential weathering of these soft and hard rocks leads to the formation of multiple sets of joint fractures (Fig 15a). Moreover, the long-term weathering and freeze-thaw cycles cause these joints to intersect and cut through the rock mass, creating unstable hazardous rock bodies (Fig 15b). These processes eventually form a continuous unloading-tension cracking zone, leading to slip-type destabilization damage (Fig 15c). Thus, the primary destabilization mode of the slope is unloading-tension cracking and slipping.

## 4. Discussion

It is evident that the freeze-thaw cycles significantly affected the stability of the rock slope, when the numerical simulation results under two different conditions were further compared. To propose an effective prevention technique to maintain the stability of the rock slopes, the modified numerical model with the application of the anchor rods was set-up [32–34]. The length of the anchor rods was 8 meters, the value of which was determined upon the maximum distance between the slope surface and the rearmost cracks of the unstable rock mass. As depicted in Figs 16 and 17, the changes of surface displacement is not obvious. The displacements along the Y-direction were only -0.58 mm and 2.68 mm, respectively. The similar

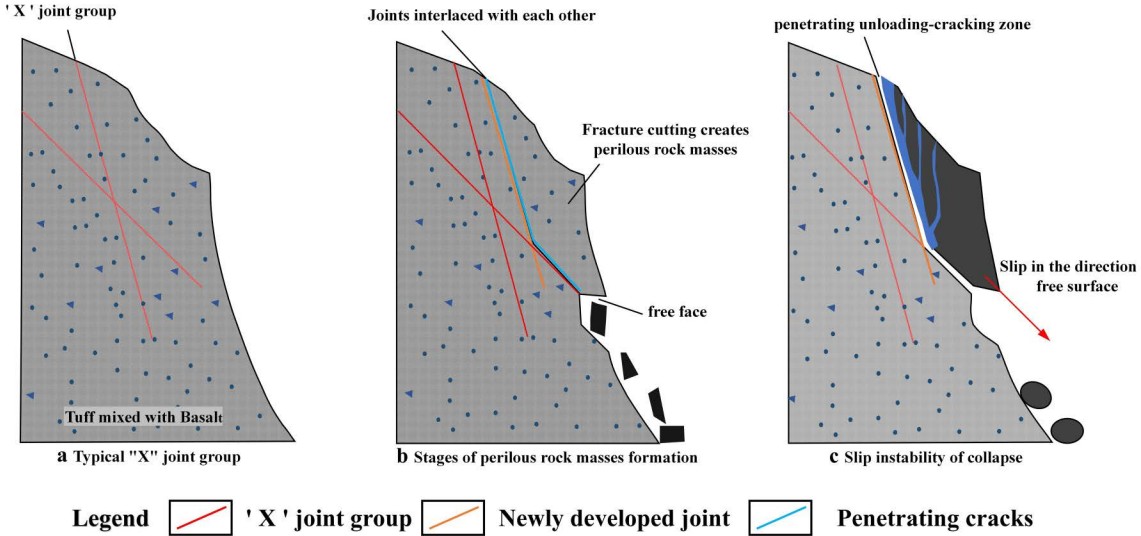

**Fig 15. Collapse failure mechanism diagram.** (a) Typical "X" joint group (b) Stages of perilous rock masses formation (c) Slip instability of collapse.

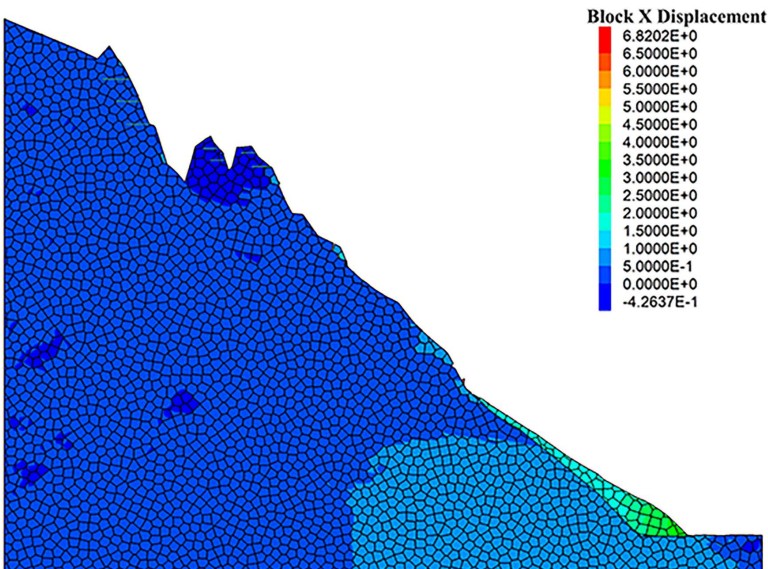

**Fig 16. X-direction displacement cloud diagram of anchored condition model.**

observation also obtained from the X-direction, the displacement of which were -0.42 mm and 3.14 mm. All these observation verified the effectiveness of the anchoring system. Meanwhile, it can be seen that the gravity-induced accumulation mainly occurred at the foot of the slope. To prevent and control the collapse of the rock slope, the application of the anchor rods seems to be feasible to help eliminate the threat of rock mass collapse and improve the geological environment of the Camel Peak area.

Figs 18 and 19 illustrate the stress cloud diagrams of the rock slope, which suggests that the monitored force remained relatively balanced without significant fluctuations from the

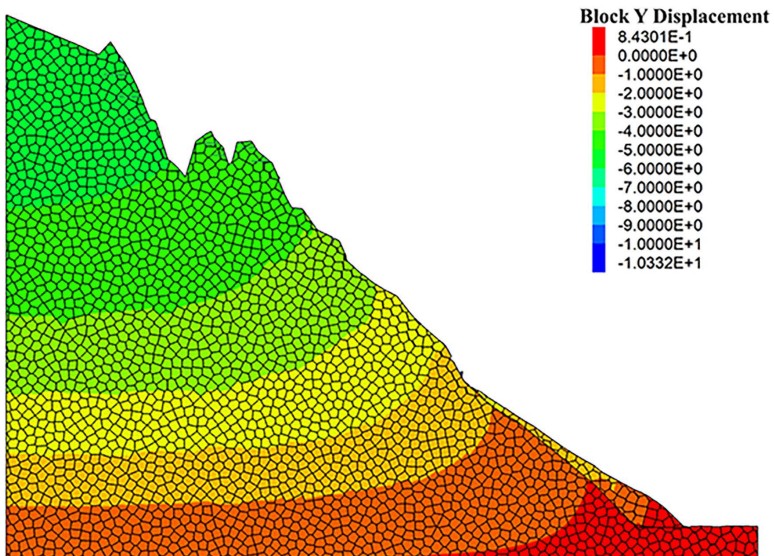

**Fig 17. Y-direction displacement cloud diagram of anchored condition modell.**

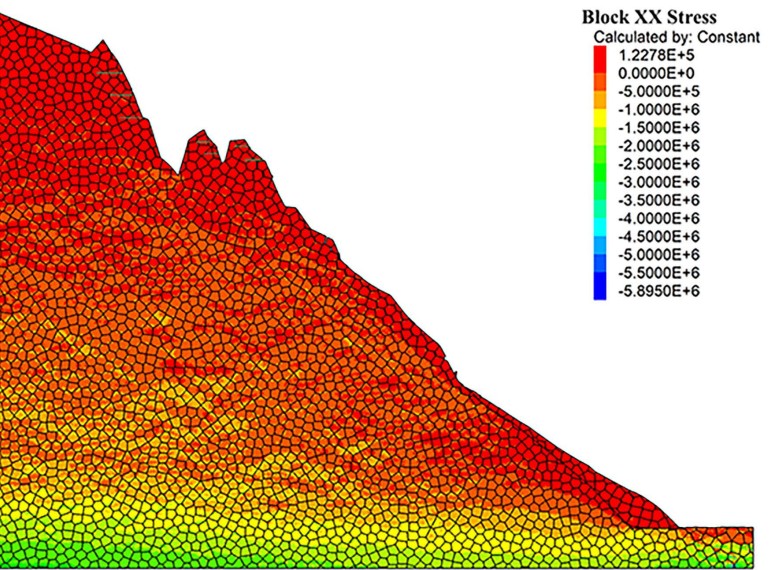

**Fig 18. X-direction stress cloud diagram of anchored condition model.**

initial stage of stress fluctuations to equilibrium. During this process, the stress changes at the top zone of the slope were relatively minor, while the increased stress at the foot of the slope was noticeable. By simulating natural freeze-thaw and bolt conditions using UDEC software, the calculation model can be monitored to assess slope stability based on observed displacement and cracks. In practical engineering applications, the application of the UDEC software allows for quicker assessment of dangerous rock mass stability in a specific area, which may facilitate risk evaluation, and enable timely prevention and control of unstable hazardous rock mass.

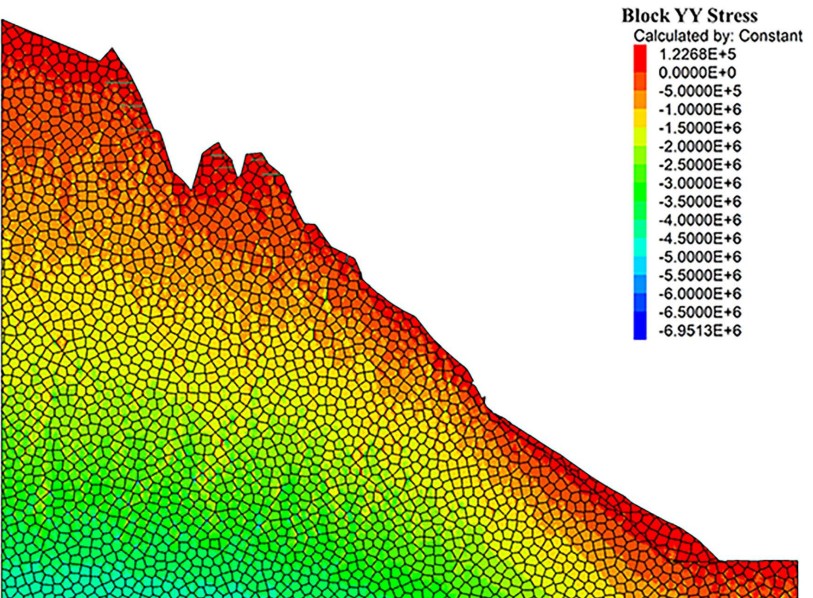

**Fig 19. Y-direction stress cloud diagram of anchored condition model.**

## 5. Conclusions

In the present research, the UDEC software was applied to investigate the failure mechanism of soft– hard- interbedded rock slopes under the natural, freeze-thaw, and anchored conditions. The following conclusions are listed for reference:

(1) The freeze-thaw cycles significantly impact the structural integrity of soft– hard- interbedded rock slopes, which is different from that under the natural conditions. Extra attention should be thus paid to at these regions with large temperature differences;

(2) Long-term weathering, freeze-thaw cycles, and other factors lead to differential weathering results in irregular unstable structural planes of the soft– hard- interbedded rock slope, leading to the instability of the soft– hard- interbedded rock slope;

(3) The failure mode of the soft– hard- interbedded rock slope is featured with the unloading, stretching, and sliding process;

(4) The damage is mainly concentrated in the surface layer of the slope and the newly exposed rock body will be re-weathering and freezing and thawing effect after the destabilization;

(5) The implementation of the anchor rods is effective in preventing and controlling the collapse of the soft– hard- interbedded rock slope;

## Supporting information

**S1 File.** **Supporting information mainly includes three parts : Monitoring data, model and UDEC software code. Monitoring data is the stress and strain monitoring point data of numerical simulation under different working conditions**. Model is the DXF format file used in this model ; UDEC software code is a command stream used under different working conditions.
(ZIP)

## Author contributions

**Data curation:** Lilong Ma, Yanyang Zhang, Qianli Lv.

**Funding acquisition:** Lilong Ma, Zizhao Zhang.

**Investigation:** Guangming Shi, Junpeng Huang.

**Methodology:** Lilong Ma, Runsen Lai.

**Supervision:** Lilong Ma, Qianli Lv.

**Visualization:** Zizhao Zhang, Yanyang Zhang.

**Writing – original draft:** Lilong Ma, Runsen Lai.

**Writing – review & editing:** Zizhao Zhang.

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
