## [Decision Letter · Decision Letter 0]

12 Aug 2024

PONE-D-24-28810

Failure Mechanisms of Soft– hard- interceded Rock Slopes in Cold Regions

PLOS ONE

Dear Dr. Zhang,

Thank you for submitting your manuscript to PLOS ONE. After careful consideration, we feel that it has merit but does not fully meet PLOS ONE’s publication criteria as it currently stands. Therefore, we invite you to submit a revised version of the manuscript that addresses the points raised during the review process.

We look forward to receiving your revised manuscript.

Kind regards,

Davood Fereidooni, Ph.D.

Academic Editor

PLOS ONE

Journal Requirements:

   "This work was supported by the following 2 funds, programs, projects: Scientific Research Programme "Tianshan Excellence":<Mechanisms of coupled geological-hazard-hydrogeological-ecological-environmental feedback of mining on the north slope of Tianshan Mountain and its engineering geological significance>(2023TSYCCX0010), The National Natural Science Foundation of China (42367021)."

5. We note that your Data Availability Statement is currently as follows: All relevant data are within the manuscript and its Supporting Information files.

6. Please amend either the title on the online submission form (via Edit Submission) or the title in the manuscript so that they are identical.

7. Please remove your figures from within your manuscript file, leaving only the individual TIFF/EPS image files, uploaded separately. These will be automatically included in the reviewers’ PDF.

8. We note that Figures 14-15 in your submission contain copyrighted images. All PLOS content is published under the Creative Commons Attribution License (CC BY 4.0), which means that the manuscript, images, and Supporting Information files will be freely available online, and any third party is permitted to access, download, copy, distribute, and use these materials in any way, even commercially, with proper attribution. For more information, see our copyright guidelines: http://journals.plos.org/plosone/s/licenses-and-copyright.

a. You may seek permission from the original copyright holder of Figures 14-15 to publish the content specifically under the CC BY 4.0 license. 

Additional Editor Comments:

Dear Dr. Zizhao Zhang,

I would like to thank you very much for submitting your manuscript to us for consideration. I have received now reviewers’ comments from our advisors on your manuscript, PONE-D-24-28810", Failure Mechanisms of Soft–hard- interceded Rock Slopes in Cold Regions". In this regard, I have come to the conclusion that your manuscript must be revised before publication in PLOS ONE. Below, please find the comments for your perusal. Please perform all required corrections based on the reviewers’ comments to the manuscript and resubmit its modified version to the journal.

With kind regards,

Davood Fereidooni, Ph.D.

Reviewers’ comments:

Reviewer#1

The authors submitted a very interesting article and in my opinion the paper should be published after minor revisions.

Comment 1: Purpose of this study is not given clearly (The authors have to indicate how to use results of this study in engineering practice). The authors should refer to novelty of the paper in the introduction section.

Comment 2: The affiliation should include city, state/province (if applicable).

Comment 3: Manuscript text should be double-spaced.

Comment 4: The text contains quite syntax and grammatical errors. As a reviewer I cannot correct all these errors. The manuscript should be carefully edited by a native English Speaker and some parts of text should be written again. I have highlighted some errors.

For example, in Line 13 you should delete “A”. Furthermore, you should use the word “interbedded” instead of “Interceded”.

In Line 18, you should delete “-'” after natural and the sentence will become “freeze-thaw cycles, natural, and anchored surroundings”

In Line 23 without “the”.

In Line 25 do not repeat the word “research”.

In Line 28: what means “UDEC” ? You should explain it.

In Line 36: “the” with capital the first letter.

In Line 38: “Promotes” instead of “promote”

Lines 45 – 46 are confused. Write them with another way.

Line 53 “are” instead of “is”.

Line 61 “make” instead of “makes”

Comment 5: In Line 114: In Table 2 uniaxial compressive strength is not shown.

Comment 6: Line 158: “monitoring point 3” instead of “monitoring point 2”

Comment 7: Line 160: “monitoring points 2 and 1” instead of “monitoring points 2 and 3”

Comment 8: Line 186: “ the forces at each monitoring point in Figure 12b increase before gradually stabilizing”. Are you sure that it increases?

Comment 9: In Lines 194-197, the sentence should be written better.

Comment 10: The references are not in the right format.

Comment 11: The figures are not in the right format.

Comment 12: The conclusion is brief in length. Write more sentences and more analytically. Also the conclusions should try to mention some form of practical application of the results. Or further research that can lead to a practical use.

Reviewer#2

The paper presents a clear and concise overview of the research on the impact of freeze-thaw on rock slopes with weak interlayers. The problem statement is well-defined and the key findings are summarized effectively. However, there are opportunities to strengthen the paper by providing more quantitative details and implications.

• The introduction should be strengthened with more recent references to support the research gap and the significance of the study. The number of references is inadequate.

• Explain more about how to measure strength values and the standard methods used for values in Tables 1 and 2.

• Has the role of ice wedge pressure in rock joints been seen for modeling?

• The density of rock materials in natural state and freezing – thawing are considered the same in modeling. This is despite the fact that the rock must face a decrease in density after enduring freezing.

• Provide more details about the numerical simulation methods.

• Discuss the implications of the findings for slope stability and risk assessment.

• Conduct a sensitivity analysis to assess the impact of key model parameters.

• Clearly articulate the practical implications of the findings.

Reviewers' comments:

Reviewer's Responses to Questions

**Comments to the Author**

1. Is the manuscript technically sound, and do the data support the conclusions?

Reviewer #1: Partly

Reviewer #2: Yes

2. Has the statistical analysis been performed appropriately and rigorously? 

Reviewer #1: Yes

Reviewer #2: I Don't Know

3. Have the authors made all data underlying the findings in their manuscript fully available?

Reviewer #1: No

Reviewer #2: Yes

4. Is the manuscript presented in an intelligible fashion and written in standard English?

Reviewer #1: No

Reviewer #2: Yes

5. Review Comments to the Author

Reviewer #1: The authors submitted a very interesting article and in my opinion the paper should be published after minor revisions.

Comment 1: Purpose of this study is not given clearly (The authors have to indicate how to use results of this study in engineering practice). The authors should refer to novelty of the paper in the introduction section.

Comment 2: The affiliation should include city, state/province (if applicable).

Comment 3: Manuscript text should be double-spaced.

Comment 4: The text contains quite syntax and grammatical errors. As a reviewer I cannot correct all these errors. The manuscript should be carefully edited by a native English Speaker and some parts of text should be written again. I have highlighted some errors.

For example, in Line 13 you should delete “A”. Furthermore, you should use the word “interbedded” instead of “Interceded”.

In Line 18, you should delete “-“ after natural and the sentence will become “freeze-thaw cycles, natural, and anchored surroundings”

In Line 23 without “the”.

In Line 25 do not repeat the word “research”.

In Line 28: what means “UDEC” ? You should explain it.

In Line 36: “the” with capital the first letter.

In Line 38: “Promotes” instead of “promote”

Lines 45 – 46 are confused. Write them with another way.

Line 53 “are” instead of “is”.

Line 61 “make” instead of “makes”

Comment 5: In Line 114: In Table 2 uniaxial compressive strength is not shown.

Comment 6: Line 158: “monitoring point 3” instead of “monitoring point 2”

Comment 7: Line 160: “monitoring points 2 and 1” instead of “monitoring points 2 and 3”

Comment 8: Line 186: “ the forces at each monitoring point in Figure 12b increase before gradually stabilizing”. Are you sure that it increases?

Comment 9: In Lines 194-197, the sentence should be written better.

Comment 10: The references are not in the right format.

Comment 11: The figures are not in the right format.

Comment 12: The conclusionς are brief in length. Write more sentences and more analytically. Also the conclusions should try to mention some form of practical application of the results. Or further research that can lead to a practical use.

Reviewer #2: The paper presents a clear and concise overview of the research on the impact of freeze-thaw on rock slopes with weak interlayers. The problem statement is well-defined and the key findings are summarized effectively. However, there are opportunities to strengthen the paper by providing more quantitative details and implications.

• The introduction should be strengthened with more recent references to support the research gap and the significance of the study. The number of references is inadequate.

• Explain more about how to measure strength values and the standard methods used for values in Tables 1 and 2.

• Has the role of ice wedge pressure in rock joints been seen for modeling?

• The density of rock materials in natural state and freezing – thawing are considered the same in modeling. This is despite the fact that the rock must face a decrease in density after enduring freezing.

• Provide more details about the numerical simulation methods.

• Discuss the implications of the findings for slope stability and risk assessment.

• Conduct a sensitivity analysis to assess the impact of key model parameters.

• Clearly articulate the practical implications of the findings.

6. PLOS authors have the option to publish the peer review history of their article (what does this mean? ). If published, this will include your full peer review and any attached files.

**Do you want your identity to be public for this peer review?** For information about this choice, including consent withdrawal, please see our Privacy Policy .

Reviewer #1: No

Reviewer #2: No

---

## [Author Response · Author response to Decision Letter 1]

4 Sep 2024

Changes have been made accordingly as requested in the e-mail.

---

## [Decision Letter · Decision Letter 1]

4 Oct 2024

PONE-D-24-28810R1Failure Mechanisms of Soft– hard- interbedded Rock Slopes in Cold Regions：Numerical Simulation and Theoretical AnalysisPLOS ONE

Dear Dr. Zhang,

Thank you for submitting your manuscript to PLOS ONE. After careful consideration, we feel that it has merit but does not fully meet PLOS ONE’s publication criteria as it currently stands. Therefore, we invite you to submit a revised version of the manuscript that addresses the points raised during the review process.

We look forward to receiving your revised manuscript.

Kind regards,

Davood Fereidooni, Ph.D.

Academic Editor

PLOS ONE

Journal Requirements:

Additional Editor Comments:

Dear Dr. Zizhao Zhang,

I would like to thank you very much for submitting your manuscript to us for consideration. I have received now reviewers’ comments from our advisors on your manuscript, PONE-D-24-28810R1", Failure Mechanisms of Soft–hard- interceded Rock Slopes in Cold Regions". In this regard, I have come to the conclusion that your manuscript must be revised before publication in PLOS ONE. Below, please find the comments for your perusal. Please perform all required corrections based on the reviewers’ comments to the manuscript and resubmit its modified version to the journal.

With kind regards,

Davood Fereidooni, Ph.D.

Academic Editor

Reviewers’ comments:

Reviewer#1

The authors submitted a very interesting article and in my opinion the paper should be published after minor revisions.

Comment 1: Despite the fact that you corrected a lot of syntax and grammar errors, there are some ones to correct.

The manuscript should be carefully edited by a native English Speaker and some parts of text should be written again.

Comment 2:In table 1 and 2, you write "pniaxiai" instead of "uniaxial"

Comment 3: In table 12, there is decrease of forces. Not increase. Otherwise, you should refer that the tensile stress increases. you should explain it better.

Comment 4: The conclusions are brief in length. Write more sentences and more analytically. Also, the conclusions should try to mention some form of practical application of the results. Or further research that can lead to a practical use.

Reviewer#2

The authors have addressed all the comments raised by the reviewers in a comprehensive and satisfactory manner. The revised manuscript is well-written, well-structured, and presents a clear and compelling argument. I am happy to recommend this article for publication.

Reviewers' comments:

Reviewer's Responses to Questions

**Comments to the Author**

1. If the authors have adequately addressed your comments raised in a previous round of review and you feel that this manuscript is now acceptable for publication, you may indicate that here to bypass the “Comments to the Author” section, enter your conflict of interest statement in the “Confidential to Editor” section, and submit your "Accept" recommendation.

Reviewer #1: All comments have been addressed

Reviewer #2: All comments have been addressed

2. Is the manuscript technically sound, and do the data support the conclusions?

Reviewer #1: Yes

Reviewer #2: Yes

3. Has the statistical analysis been performed appropriately and rigorously? 

Reviewer #1: Yes

Reviewer #2: Yes

4. Have the authors made all data underlying the findings in their manuscript fully available?

Reviewer #1: Yes

Reviewer #2: Yes

5. Is the manuscript presented in an intelligible fashion and written in standard English?

Reviewer #1: Yes

Reviewer #2: Yes

6. Review Comments to the Author

Reviewer #1: The authors submitted a very interesting article and in my opinion the paper should be published after minor revisions.

Comment 1: Despite the fact that you corrected a lot of syntax and grammar errors, there are some ones to correct.

The manuscript should be carefully edited by a native English Speaker and some parts of text should be written again.

Comment 2:In table 1 and 2, you write "pniaxiai" instead of "uniaxial"

Comment 3: In table 12, there is decrease of forces. Not increase. Otherwise, you should refer that the tensile stress increases. you should explain it better.

Comment 4:The conclusionς are brief in length. Write more sentences and more analytically. Also the conclusions should try to mention some form of practical application of the results. Or further research that can lead to a practical use.

Reviewer #2: The authors have addressed all the comments raised by the reviewers in a comprehensive and satisfactory manner. The revised manuscript is well-written, well-structured, and presents a clear and compelling argument. I am happy to recommend this article for publication.

7. PLOS authors have the option to publish the peer review history of their article (what does this mean? ). If published, this will include your full peer review and any attached files.

**Do you want your identity to be public for this peer review?** For information about this choice, including consent withdrawal, please see our Privacy Policy .

Reviewer #1: No

Reviewer #2: No

---

## [Author Response · Author response to Decision Letter 2]

22 Oct 2024

The pictures without legends are supplemented.Added the name of the figure to the article.

---

## [Editor Report · Decision Letter 2]

29 Nov 2024

PONE-D-24-28810R2Failure Mechanisms of Soft– hard- interbedded Rock Slopes in Cold Regions：Numerical Simulation and Theoretical AnalysisPLOS ONE

Dear Dr. Zhang,

Thank you for submitting your manuscript to PLOS ONE. After careful consideration, we feel that it has merit but does not fully meet PLOS ONE’s publication criteria as it currently stands. Therefore, we invite you to submit a revised version of the manuscript that addresses the points raised during the review process.

We look forward to receiving your revised manuscript.

Kind regards,

Davood Fereidooni, Ph.D.

Academic Editor

PLOS ONE

Journal Requirements:

Additional Editor Comments:

Dear Author(s)

I would like to thank you very much for submitting your manuscript to us for consideration. I have received now reviewers’ comments from our advisors on your manuscript, PONE-D-24-28810R1", Failure Mechanisms of Soft–hard- interceded Rock Slopes in Cold Regions". In this regard, I have come to the conclusion that your manuscript must be revised before publication in PLOS ONE. Below, please find the reviewer comments I believe still need to be addressed from the previous round of revisions for your perusal. Please perform all required corrections based on the reviewers’ comments to the manuscript and resubmit its modified version to the journal.

With kind regards,

Davood Fereidooni, Ph.D.

Reviewer#1 Comments:

The authors submitted a very interesting article and in my opinion the paper should be published after minor revisions.

Comment 1: Despite the fact that you corrected a lot of syntax and grammar errors, there are some ones to correct.

The manuscript should be carefully edited by a native English Speaker and some parts of text should be written again.

Comment 2:In table 1 and 2, you write "pniaxiai" instead of "uniaxial"

Comment 3: In table 12, there is decrease of forces. Not increase. Otherwise, you should refer that the tensile stress increases. you should explain it better.

Comment 4:The conclusionς are brief in length. Write more sentences and more analytically. Also the conclusions should try to mention some form of practical application of the results. Or further research that can lead to a practical use.

Reviewer#2 Comments:

The authors have addressed all the comments raised by the reviewers in a comprehensive and satisfactory manner. The revised manuscript is well-written, well-structured, and presents a clear and compelling argument. I am happy to recommend this article for publication.

---

## [Author Response · Author response to Decision Letter 3]

7 Dec 2024

The opinions of reviewer 1 were further revised.

---

## [Editor Report · Decision Letter 3]

26 Dec 2024

Failure Mechanism of Soft– hard- interbedded Rock Slopes in Cold Regions：Numerical Simulation and Theoretical Analysis

PONE-D-24-28810R3

Dear Dr. Zhang,

We’re pleased to inform you that your manuscript has been judged scientifically suitable for publication and will be formally accepted for publication once it meets all outstanding technical requirements.

Kind regards,

Davood Fereidooni, Ph.D.

Academic Editor

PLOS ONE
---

## [Editor Report · Acceptance letter]

PONE-D-24-28810R3

PLOS ONE

Dear Dr. Zhang,

I'm pleased to inform you that your manuscript has been deemed suitable for publication in PLOS ONE. Congratulations! Your manuscript is now being handed over to our production team.

Kind regards,

on behalf of

Dr. Davood Fereidooni

Academic Editor

PLOS ONE